# Ambiguities in helical reconstruction

Edward H Egelman*

Department of Biochemistry and Molecular Genetics, University of Virginia, Charlottesville, United States

**Abstract** Helical polymers are found throughout biology and account for a substantial fraction of the protein in a cell. These filaments are very attractive for three-dimensional reconstruction from electron micrographs due to the fact that projections of these filaments show many different views of identical subunits in identical environments. However, ambiguities exist in defining the symmetry of a helical filament when one has limited resolution, and mistakes can be made. Until one reaches a near-atomic level of resolution, there are not necessarily reality-checks that can distinguish between correct and incorrect solutions. A recent paper in eLife (Xu et al., 2014) almost certainly imposed an incorrect helical symmetry and this can be seen using filament images posted by Xu et al. A comparison between the atomic model proposed and the published three-dimensional reconstruction should have suggested that an incorrect solution was found.

## Main text

Helical polymers are ubiquitous in biology, and are found extensively in viruses (*Ge and Zhou, 2011*), bacteria (*Polka et al., 2009*; *Ozyamak et al., 2013*), eukaryotes (*Galkin et al., 2010*) and archaea (*Yu et al., 2012*). Because of the symmetry inherent in helical filaments (where identical subunits are rotated and translated along a filament axis) projections of helical filaments can provide all of the information needed to generate a three-dimensional reconstruction from a limited number of images, and this property was exploited in the first electron microscopic (EM) three dimensional reconstruction, which was made from a helical bacteriophage tail (*DeRosier and Klug, 1968*). New advances in both computational approaches (*Egelman, 2000*) and cryo-imaging with direct electron detectors (*Bai et al., 2013*) have meant that the determination of structures of helical polymers at near-atomic resolution (*Lu et al., 2014*; *Wu et al., 2014*) is becoming much more common. However, the assignment of the correct symmetry to a helical polymer can still be problematic (*Egelman, 2010*; *Desfosses et al., 2014*), and examples exist where the wrong symmetry has been mistakenly imposed (*Okorokov et al., 2010*; *Sen et al., 2010*; *Yu and Egelman, 2010*; *Heymann et al., 2013*). Insufficient understanding exists about how solutions (three-dimensional structures) may be obtained that are stable under iterative refinement, consistent with the images, but simply wrong. This arises from imposing a helical symmetry that is incorrect, but the available resolution does not allow one to distinguish between wrong symmetries and the correct one. This short communication seeks to reconcile the very different reconstructions of the MAVS filament by two groups (*Wu et al., 2014*; *Xu et al., 2014*) and in the process raise some general issues about helical reconstruction. While the points may appear to be quite technical to the non-specialist, they raise important questions about the expertise needed to evaluate some cryo-EM results.

The reconstruction of the MAVS filament presented in Wu et al. had a stated resolution of 3.6 Å and had a helical symmetry of a rotation of −101.1° (the negative rotation corresponding to a left-handed helix) and a rise of 5.1 Å per subunit. In contrast, the reconstruction of Xu et al. had a stated resolution of 9.6 Å, a C3 point group rotational symmetry, with a rotation of −53.6° and a rise of 16.8 Å for every ring of three subunits. One of the differences between the two reconstructions is that in Wu et al. not only are α-helices clearly resolved, but bulky aromatic side-chains can be seen that are consistent with

*For correspondence: egelman@
virginia.edu

**Competing interests:** The author declares that no competing interests exist.

**Reviewing editor**: Wesley I Sundquist, University of Utah, United States

not only the sequence but a crystal structure of the subunit which can be fit quite well as a rigid body into the reconstruction. In the Comment appended to their paper (*Jiang, 2014*), Xu et al. do not dispute the validity of the reconstruction in Wu et al., but argue that the filament being reconstructed by Wu et al. is an artifact of harsh preparative procedures, and that their filaments have a different symmetry and represent a more native conformation. They advance three arguments for why their filaments have a different symmetry. Given that the authors of Xu et al. have deposited the images that they used in the EMPIAR database, these arguments can be directly tested.

Their first argument for a different helical symmetry is that a layer line at ~1/(17 Å) in their power spectrum has intensity on the meridian, while the corresponding layer line in Wu et al. does not and is clearly arising from an n = −1 Bessel order. *Figure 1* shows how the power spectrum of an image is the central section of the three-dimensional power spectrum of the object being projected onto the image. As such, the projection is sensitive to any tilt of the axis of the object out of the plane of projection. An untilted central section (*Figure 1B*) would generate the two separated peaks on both sides of the meridian seen in *Figure 1D* (arrow), while a sufficiently tilted central section (*Figure 1C*) would generate a single intensity on the meridian (*Figure 1E*). For negatively stained polymers, where filaments are adsorbed to a carbon film, out-of-plane tilt can be largely ignored. In cryo-EM, where filaments are imaged within an ice film of finite thickness, such tilt cannot be ignored, particularly as one becomes more sensitive to it the higher the resolution. The first question I asked is whether there was any substantial out-of-plane tilt for the filaments imaged by Xu et al. I used an approach of providing references with different tilt angles to generate the histogram shown in *Figure 2A*. The large peaks at −20° and +20° correspond to segments with tilt angles greater than 20°, showing the unexpectedly large degree of tilt for these filaments. If one only takes the segments in the three central bins (−4°, 0°, +4°) a power spectrum is generated (*Figure 2B*) that clearly shows (arrow) an n = −1 layer line with no intensity on the meridian. This power spectrum looks to be the same as that shown by Wu et al., undercutting the argument that the filaments being used by the two groups are different. One can calculate the tilt that would be needed to make the two separated peaks in *Figure 2B* appear as a single peak on the meridian, and it is ~9°, entirely consistent with the histogram in *Figure 2A*.

Is it possible that the sorting I have done for out-of-plane tilt is completely wrong, since it has involved the application of a helical symmetry, and the power spectrum in *Figure 2B* is actually from a highly tilted subset, while the power spectrum in Xu et al. is from an untilted subset? The answer is no, since we can see from *Figure 1* how the 2D power spectrum will change with tilt. The two separated peaks seen in *Figure 2B* would also need to be present as secondary maxima in the power spectrum shown in Xu et al. to appear as a result of tilt, and they simply are not present there. On the other hand, the power spectrum in *Figure 2B*, if assumed to be representative of filaments with little tilt, can explain the power spectrum of Xu et al. when that is assumed to represent a large tilt.

The second argument advanced by Xu et al. in their Comment is that the symmetry used by Wu et al. is unstable in the Iterative Helical Real Space Reconstruction (IHRSR) approach when applied to their filaments, and therefore their filaments must have a different symmetry. Since I developed the IHRSR method (*Egelman, 2000*), I have some experience with the application of the algorithm to many helical systems. I have argued (*Egelman, 2007*, *2010*) that a stable solution in IHRSR is a necessary, but not sufficient, requirement for the solution to be correct, since it has been clear that many wrong solutions can also be stable. However, it has become apparent in cryo-EM, when out-of-plane tilt cannot be ignored, that correct solutions can be unstable when this tilt is ignored. *Figure 3* shows the symmetry parameters (rotation and axial rise per subunit) for the filaments of Xu et al. when references are included with out-of-plane tilt. It can be seen that the symmetry determined by Wu et al. can be applied to the filaments of Xu et al. and that this is quite stable in IHRSR cycles, again undercutting their argument that the two sets of filaments are different.

The third and last argument advanced by Xu et al. in their Comment is that their reconstruction has a hole in the center, while the model of Wu et al. that has been built into the 3.6 Å resolution reconstruction would have no such hole. They correctly state that such a hole can be seen in their filaments independently of the symmetry simply by cylindrically averaging their 3D density map. If one takes the cryo-EM density map deposited by Wu et al. (EMDB-5922) and filters it to 12 Å resolution a hole is seen in the center (*Figure 4B*, arrow). The reason that this is consistent with the deposited model (3j6j.PDB) is simply that the subunits are helically arranged and do not pack to form a solid core. Since the packing of protein is less dense in the center of the filament than it is at higher radius, a cylindrically averaged density distribution will show a continuous hole in the center of the filament. The reason

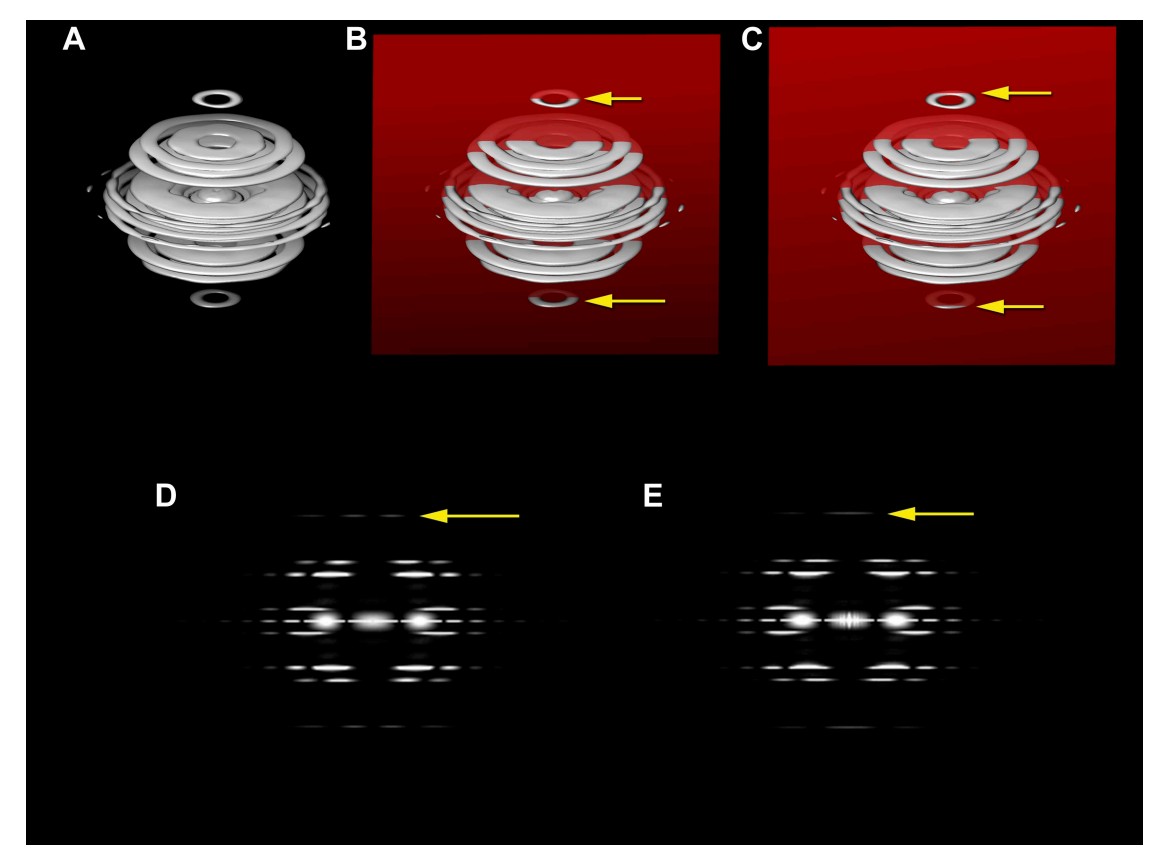

**Figure 1**. The three-dimensional power spectrum of a helical filament and power spectra from projections of the filament. The three-dimensional power spectrum of a helical filament is shown (**A**). The power spectrum from a cryo-EM image of a helical filament (where the image is a projection of a three-dimensional object onto a two-dimensional plane) is the central section of the three-dimensional power spectrum, and this would be given by the intersection of the red plane in (**B**) with the power spectrum. The yellow arrows in (**B**) show the intersection of this plane with a layer plane that arises from a 1-start helix. The intensities on this central section are shown in (**D**), which would be the power spectrum from untilted filaments, i.e., those whose filament axis is parallel to the plane of projection. The yellow arrow in (**D**) shows the layer line from the 1-start helix, with peak intensities on both sides of the meridian. Now consider what happens when the filament axis is tilted away from parallel to the plane of projection. The central section (**C**) would now intersect the 1-start layer plane at the positions given by the two yellow arrows, and in the two-dimensional power spectrum the two peaks would collapse to a single peak on the meridian (**E**, yellow arrow). If one did not know that the filament was tilted, such intensity would be mistaken for a meridional layer line with the Bessel order n = 0.

that I have chosen 12 Å for this comparison is that this is the actual resolution of the reconstruction (*Figure 4A,C*) that I have been able to generate from the images of Xu et al. Why is the resolution so poor? The most obvious differences are that the Xu et al. images appear to be from thick ice, there is a very noisy background, and they have applied a carbon film which degrades the signal-to-noise ratio. Since they are sampling at 2.3 Å/px, the best one might hope to achieve would be ~7 Å resolution (2/3 Nyquist) but given the problems with the images one would never get even close to this.

I have thus shown that the three arguments for why their filaments are different from those of Wu et al., advanced by Xu et al. in their Comment, are not true. This does not establish, however, that the filaments are the same. Establishing that would require a high resolution reconstruction from the filaments of Xu et al. which I fear would be impossible. But we can look at their published reconstruction, which gets to the most important point: given all of the potential problems and ambiguities in helical reconstruction, how does one know that a structure is correct? The great success of protein x-ray crystallography lies in the fact that structures are being solved that all have a known stereochemistry. One has prior knowledge about what an α-helix and a β-sheet look like, as well as knowing the amino acid sequence that must be built into a model. At lower resolution by EM most of these reality-checks are simply absent, and one needs to generate methods, such as the tilt-pair validation used for

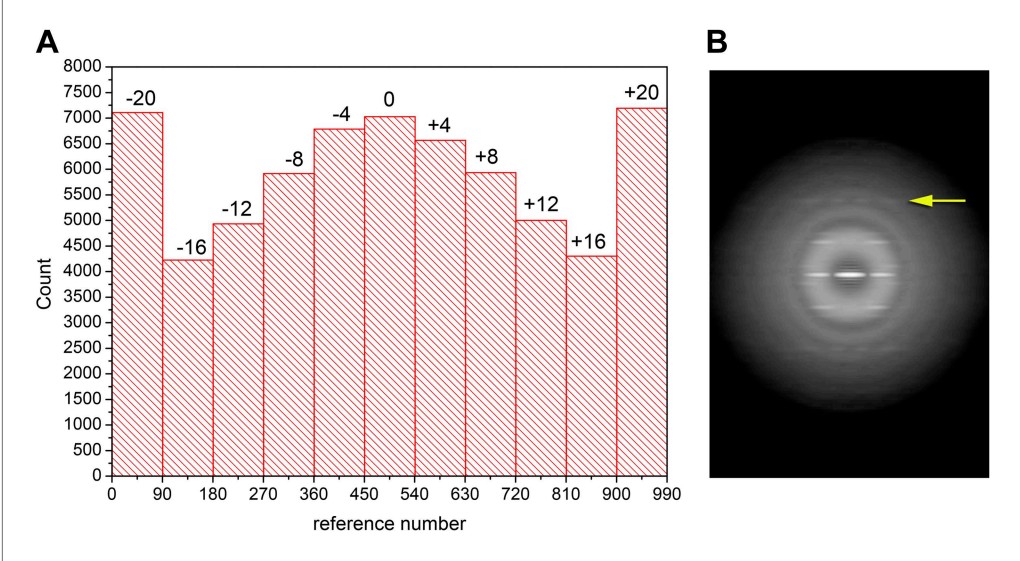

**Figure 2**. Out-of-plane tilt cannot be ignored in the images of Xu et al. A histogram (**A**) of the out-of-plane tilt seen in the filaments of Xu et al. A reconstruction was generated from these filaments while allowing for out-of-plane tilt, and this reconstruction was used as a reference, generating 90 different azimuthal projections (4° increments) for tilt angles from −20° to +20° (also with 4° increments), generating 990 reference projections. The large peaks at both ends of the distribution arise from truncating the search range to smaller values than actually found in the filament population. Using filament segments in the three central bins (−4°, 0°, 4°) for generating a power spectrum (**B**) it can be seen that there is no meridional intensity on the layer line indicated by the arrow at ~1/(18 Å), and the appearance of this layer line is now the same as in Wu et al. The log of the power spectrum is shown to better display the large dynamic range.

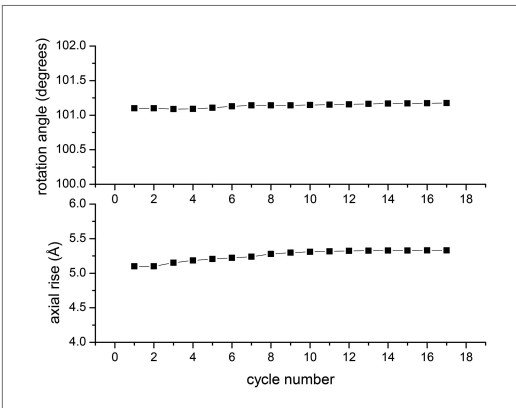

**Figure 3**. In the IHRSR method, the helical parameters (rotation and rise per subunit) are refined each cycle. The starting parameters were the symmetry determined by Wu et al., and it can be seen that when using the images of Xu et al. these parameters are perfectly stable. The first seven cycles were run with images decimated from 2.3 Å/px to 4.6 Å/px, ignoring out-of-plane tilt. The subsequent cycles were run with the undecimated images and allowing for out-of-plane tilt.

single-particle cryo-EM (*Rosenthal and Henderson, 2003*) that can distinguish between correct and incorrect reconstructions. As the field of cryo-EM is now rapidly moving to near-atomic resolution, seeing recognizable secondary structure provides a necessary and sufficient validation of a reconstruction. In the case of Wu et al. having a map with near-atomic detail, showing right-handed α-helices, confirmed not only the symmetry but the absolute hand of the reconstruction, whereas in the absence of this level of detail one must use means such as metal-shadowing or tomography to resolve the enantiomorphic ambiguity present. But at lower resolution one can still ask how well a reconstruction agrees with a model. Unfortunately, there is vanishingly little agreement between their deposited map (EMDB-5890) and their deposited model (PDB 3j6c).

I show a comparison between their actual reconstruction (*Figure 5A,C*), and their model filtered to 9.6 Å (*Figure 5B,D*), the stated resolution of their reconstruction. One sees at these higher thresholds than used in Xu et al. that there is actually no match between the features in one and in the other. A reasonable question is whether such comparisons are fair, since the atomic models are free from any noise, while the reconstruction may contain noise at the resolution limit. Several published examples suggest that this method is

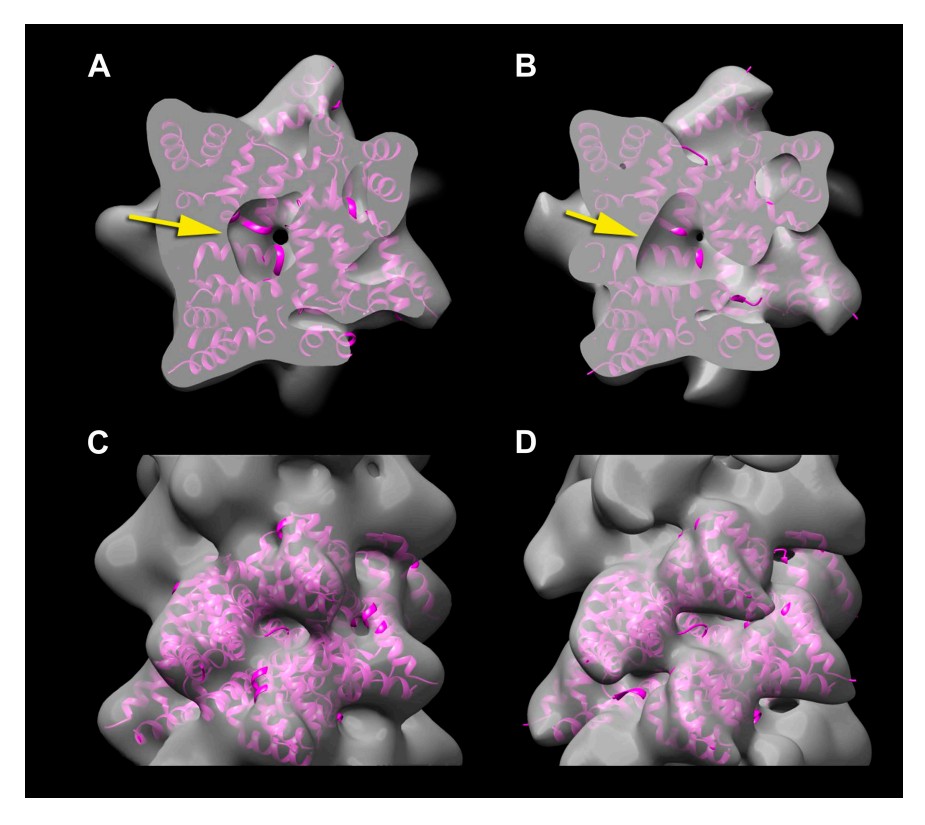

**Figure 4**. A reconstruction has been made from the filaments of Xu et al. (**A** and **C**) by applying the helical symmetry described in Wu et al. For purposes of comparison, the reconstruction of Wu et al. (EMDB-5922) has been filtered to 12 Å resolution (**B** and **D**). It can be seen that with similar thresholds, both reconstructions show a hole in the center (**A** and **B**, arrows), arising from the helical procession of the subunits about the filament axis. The atomic model (3J6J.PDB) from *Wu et al. (2014)* is shown fit into both reconstructions.

reasonable, and might be the best gauge of the actual reconstruction. In a 9 Å resolution reconstruction of an actin-cofilin filament a comparison with an atomic model filtered to this resolution (Supplementary figure 1D, *Galkin et al., 2011*) shows an excellent match. Similarly, a 7.5 Å resolution reconstruction of a naked actin filament shows an excellent match with an atomic model filtered to this resolution (Supplementary figure 2, *Fujii et al., 2010*).

This comparison between a map and a model can be made quantitatively, and *Figure 5E* shows a Fourier Shell Correlation (FSC) plot between the map and model of Xu et al., with a resolution estimate of ~22 Å. Such a resolution suggests that the only agreement between the map and the model is for compact 'blobs' (which is what this small subunit looks like at a resolution of 22 Å) related by the same imposed helical symmetry. This type of comparison, had it been done by the authors, should have raised many questions. One is whether the helical symmetry imposed was actually correct. Another is whether the FSC, which is only a measure of internal consistency between two halves of the image set and not a measure of actual resolution, has any meaning in this instance since it yields such a different result between two reconstructions (each generated from half of the image set) and between the overall reconstruction and the model.

In conclusion, the arguments of Xu et al. for a different symmetry in their filaments than that in Wu et al. collapse completely when one actually examines the publicly deposited data.

## Materials and methods

The micrographs in set04 within the EMPIAR-10014 deposition (http://pdbe.org/empiar) from *Xu et al. (2014)* were used for analysis. Of the 581 images, the first 160 were used for filament extraction.

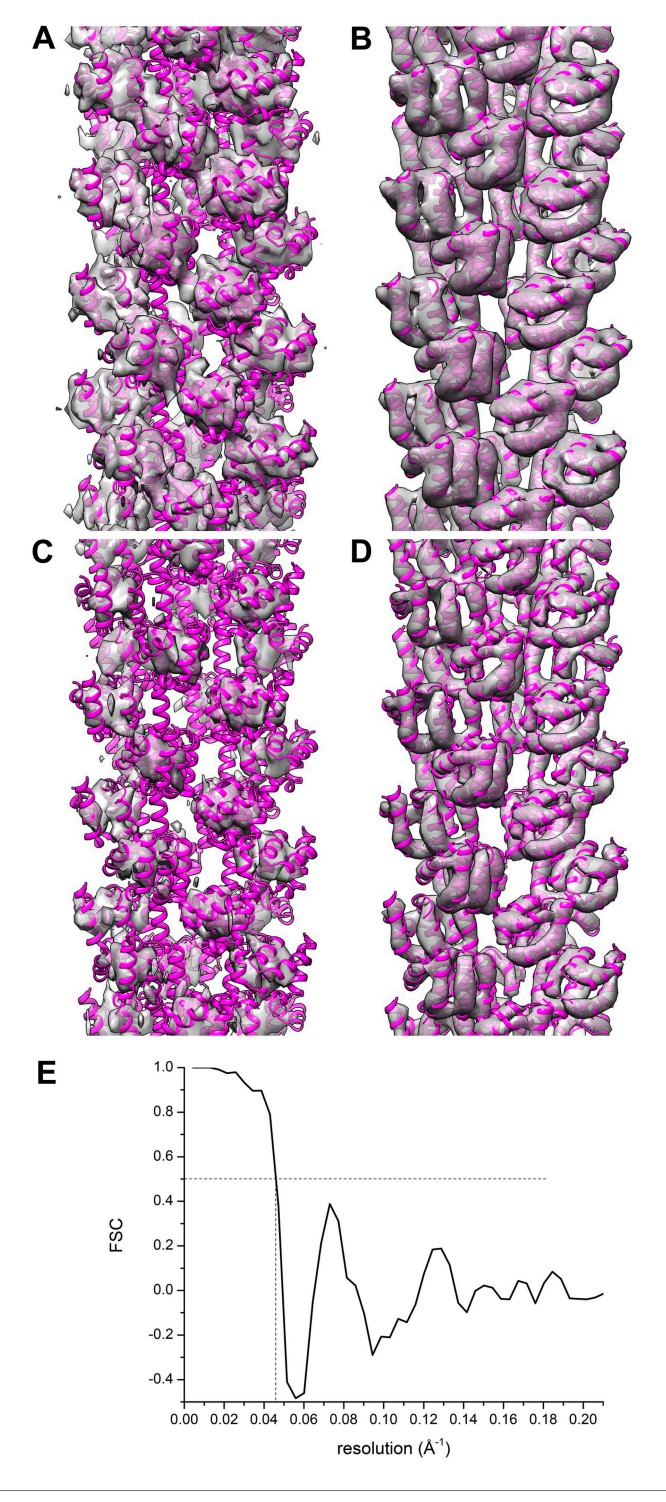

**Figure 5**. Comparisons between a model and the actual map are informative. The reconstruction (**A** and **C**) of Xu et al. (EMDB-5890) is compared with their model 3j6c.PDB (**B** and **D**). In all four panels (**A**–**D**) the same model (3j6c.PDB) is shown as magenta ribbons, but the transparent surfaces in **B** and **D** have been generated by filtering the model density to 9.6 Å, the stated resolution of the reconstruction shown in (**A** and **C**). The surfaces in (**C** and **D**) are shown at a higher threshold than in (**A** and **B**). It can be seen that with these thresholds, there is virtually no correlation between the map (**A** and **C**) and the model (**B** and **D**). A Fourier Shell Correlation between their model and reconstruction (**E**) shows that at FSC = 0.5 the resolution is ~22 Å, which is consistent with the visual comparison of the map and model (**A**–**D**).

The Contrast Transfer Function was measured using CTFFIND3 (*Mindell and Grigorieff, 2003*). For the 160 images, the mean defocus was 3.2 μ and the range was from 1.7 μ to 5.4 μ. The images were then multiplied by the theoretical CTF function to correct the phases and improve the SNR. From these 160 images, 1066 long filament boxes were cut using the e2helixboxer routine within EMAN2 (*Tang et al., 2007*). The SPIDER software package (*Frank et al., 1996*) was used for all further steps. Overlapping boxes 192 px long (2.3 Å/px) were cut from the long filament boxes using a shift of 6 px (97% overlap), generating 64,980 boxes. These were used for IHRSR (*Egelman, 2000*) which converged to a symmetry of −101.2° rotation with an axial rise of 5.3 Å, starting from a solid cylinder as an initial reference. Due to the very poor resolution (~17 Å), an attempt was made to improve this by only using the 58 images with a defocus less than 3.0 μ, which yielded 15,624 overlapping boxes. Because segments with a very large out-of-plane tilt were excluded, the final reconstruction was generated from 8600 segments. The modulation of the amplitudes of the final reconstruction by the CTF (the phase reversals of the CTF were previously corrected when the images were multiplied by the CTF) was corrected by dividing the Fourier coefficients in the reconstruction by the sum of the squared CTFs, imposing a negative B-factor of 2000 to correct for the decay of the high frequencies, and filtering to 12 Å.

## Acknowledgements

This work has been supported by NIH EB001567. I thank Albina Orlova for assistance with particle picking.

## Additional information

### Funding

| Funder | Grant reference number | Author |
|---|---|---|
| National Institutes of Health | R01 EB001567 | Edward H Egelman |

The funder had no role in study design, data collection and interpretation, or the decision to submit the work for publication.

### Author contributions

EHE, Conception and design, Analysis and interpretation of data, Drafting or revising the article

## Additional files

### Major dataset

The following previously published datasets were used:

| Author(s) | Year | Dataset title | Dataset ID and/or URL | Database, license, and accessibility information |
|---|---|---|---|---|
| Xu H, He X, Zheng H, Huang L, Hou F, Yu Z, de la Cruz MJ, Borkowski B, Zhang X, Chen ZJ, Jiang QX | 2014 | MAVS CARD | EMPIAR-10014 | Publicly available at the Electron Microscopy Pilot Image Archive (http://www.ebi.ac.uk/pdbe/emdb/empiar/). |
| Wu B, Peisley A, Li Z, Egelman E, Walz T, Penczek P, Hur S | 2014 | 3.6 Angstrom resolution MAVS filament generated from helical reconstruction | http://www.ebi.ac.uk/pdbe/entry/EMD-5922 | Publicly available at the Electron Microscopy Data Bank. |
| Wu B, Peisley A, Tetrault D, Li Z, Egelman EH, Magor KE, Walz T, Penczek PA, Hur S | 2014 | 3.6 Angstrom resolution MAVS filament generated from helical reconstruction | http://www.rcsb.org/pdb/explore/explore.do?structureId=3j6j | Publicly available at RCSB Protein Data Bank. |
| Xu H, He X, Zheng H, Huang LJ, Hou F, Yu Z, de la Cruz MJ, Borkowski B, Zhang X, Chen ZJ, Jiang QX | 2014 | Cryo-EM structure of MAVS CARD filament | http://www.ebi.ac.uk/pdbe/entry/EMD-5890 | Publicly available at the Electron Microscopy Data Bank. |

| Xu H, He X, Zheng H, Huang LJ, Hou F, Yu Z, de la Cruz MJ, Borkowski B, Zhang X, Chen ZJ, Jiang QX | 2014 | Cryo-EM structure of MAVS CARD filament | http://www.rcsb.org/pdb/explore/explore.do?structureId=3j6c | Publicly available at RCSB Protein Data Bank. |

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
