## [Decision Letter]

Thank you for sending your work entitled “Ambiguities in Helical
Reconstruction” for consideration at *eLife*. Your article has
been favorably evaluated by John Kuriyan (Senior editor), Wes Sundquist (Reviewing
editor), and three reviewers, of whom Niko Grigorieff and Carsten Sachse have agreed to
reveal their identity. A third reviewer remains anonymous.

The Reviewing editor and the three reviewers discussed their comments before we reached
this decision, and the Reviewing editor has assembled the following comments to help you
prepare a revised submission.

Dr Egelman analyzes the validity of a cryoEM helical reconstruction of the prion-like
filament formed by the CARD domain of the RIG-I adaptor molecule MAVS, as reported by Xu
et al. in *eLife* (Xu et al. *eLife*, 2014, 3:e01489). The
study by Xu et al. was published before a related study was published by Wu et al. in
*Molecular Cell* (Wu et al. *Molecular Cell* 2014,
55:511-524). The situation merits follow-up because the two reported MAVS structures are
entirely different. There is no dispute that the Wu et al. study contained high quality
EM data, that their reconstruction was technically correct, and that they successfully
generated a near-atomic resolution structure of their MAVS filaments. The central
question is whether differences in sample preparation generated distinct MAVS filaments
with different physical structures, or whether the differences in the reported
structures instead reflect methodological problems in the Xu et al study. Dr Jiang (one
of the corresponding authors on the Xu et al. paper) has posted a Comment that
accompanies the original *eLife* manuscript in which he argues that the
reported differences in the filament helical symmetries and structures reflect genuine
physical differences in the MAVS filaments generated by the two groups. Here, Egelman
analyzes the central claims of that Comment and also discusses the validity of the
original Xu et al. reconstruction.

Egelman downloaded part of the original data deposited by Xu et al. in the EMPIAR data
base and performed image analyses to test the arguments put forth by Jiang that the
filaments used in the two studies were indeed different. He shows convincingly that the
three central pieces of evidence presented in the Jiang Comment are invalid. He makes
several important points, including that Xu et al. did not appreciate that their
filaments had significant out-of-plane tilt and that convergence stability in a symmetry
search is not sufficient to establish the validity of that symmetry. Importantly, he
also shows that the Xu et al. structure does not agree with its own PDB model beyond a
resolution of ∼20 Å, suggesting that the computed EM density is not correct.
These arguments convincingly rebut the points raised in Jiang's Comment, reveal
potential inconsistencies in the processing procedure used by Xu et al., and raise
substantial doubts about the validity of the Xu et al. reconstruction. These analyses
are therefore an important contribution.

The Egelman analysis could be strengthened even further by providing additional evidence
that Xu et al. must have imposed the wrong helical symmetry. The author could do this by
computing a structure with the symmetry parameters from Wu et al. imposed on the EMPIAR
data set from Xu et al. (which he has already been analyzing). Such a structure should
contain clearer densities and recognizable secondary structure features.

---

## [Author Response]

*[…] The Egelman analysis could be strengthened even further by providing
additional evidence that Xu et al. must have imposed the wrong helical symmetry. The
author could do this by computing a structure with the symmetry parameters from Wu et
al. imposed on the EMPIAR data set from Xu et al. (which he has already been
analyzing). Such a structure should contain clearer densities and recognizable
secondary structure features*.

I have been able to improve the resolution that I initially obtained with their images
by excluding those with a defocus greater than 3.0μ. This reduced the data set
from ∼64k segments to ∼15k segments, but the reconstruction is clearly
improved. I have expanded Figure 4 to show a
better comparison of the reconstruction from Xu et al. with the correct symmetry with
the one from Wu et al. filtered to 12 Å resolution, and shown the atomic model of
Wu et al. fit into both volumes.